# A Brown Bear’s Days in Vilnius, the Capital of Lithuania

**DOI:** 10.3390/ani15142151

**Published:** 2025-07-21

**Authors:** Linas Balčiauskas, Laima Balčiauskienė

**Affiliations:** State Scientific Research Institute Nature Research Centre, Akademijos 2, 08412 Vilnius, Lithuania

**Keywords:** *Ursus arctos*, Lithuania, population recovery, urban carnivores, institutional responsibilities

## Abstract

In June 2025, a two-year-old female brown bear (*Ursus arctos*) appeared in Vilnius, Lithuania, attracting widespread public and media attention. This paper explores how the incident reflected not only a wildlife management issue, but also social, emotional, and symbolic responses. The media portrayed the bear as both a threat and a spectacle, using the story to capture public interest and provide distraction during times of social stress. Although authorities issued a permit to kill the bear if necessary, local hunters refused, favoring a nonlethal approach. Monitoring the animal with drones marked a shift toward technological solutions in urban wildlife management. The bear eventually left the city peacefully, and the case was viewed as a success in nonlethal conflict resolution. Overall, this event illustrates the questions about coexistence, ethics, and the understanding and representation of nature in modern society that arise when large carnivores are present in cities.

## 1. Introduction

In mid-June 2025, a brown bear (*Ursus arctos*) visited the capital city of Lithuania, causing chaos in institutional responses [1,2]. Given that these large carnivores have only recently begun reappearing after sporadic visits over the last few decades [3], this case merits scientific analysis to inform management implications.

### 1.1. Urban and Peri-Urban Bears in the World

Cities are some of the most challenging environments for carnivorous mammals due to habitat fragmentation, limited natural resources, and altered climates. Nevertheless, even large carnivores, such as bears, wolves, and hyenas, increasingly exploit the edges of cities for food and shelter [4].

American black bears (*Ursus americanus*) have been reported in urban areas of North America [5], while *U. arctos* began foraging in some European towns decades ago [6]. As the spatial overlap between humans and bears increases, conservation efforts and scientific research are shifting toward supporting coexistence [7]. However, the urbanization of habitats is not advantageous for bears [8]. For instance, urban female *U. americanus* have higher reproductive rates but also higher mortality rates. This makes these populations demographic sinks, unable to sustain or repopulate wildland areas [5].

All bear species are considered part of the megafauna and are flagship species [9,10], which makes them attractive to tourists around the world [11]. In the Arctic, the species of interest is the polar bear (*Ursus maritimus*) [12]. In North America, the species of interest is *U. americanus*, and in Europe, it is *U. arctos* [13,14,15].

Meanwhile, the media and scientific literature have reported an increase in attacks by large carnivores on humans in North America and Europe. These incidents are often sensationalized, which contributes to public fear and negative perceptions of carnivore conservation [16]. Notably, increased human activity in carnivore habitats and frequent engagement in risk-enhancing behaviors have been identified as key drivers of many attacks. Approximately half of the well-documented cases are linked to such behavior. Reports of *U. arctos* aggression toward humans span North America [17] and several European countries [18,19,20,21], emphasizing the urgent need for context-specific, evidence-based strategies to reduce conflict and foster coexistence.

Human–wildlife conflicts often involve large, bold carnivores with reduced fear of humans, increasing the risk for people and animals alike. These individuals play a significant role in shaping public attitudes and management responses. Studies in Slovenia and Italy revealed that bold bears were more active during the day and used open areas more frequently than cautious bears [22].

### 1.2. Brown-Bear–Human Conflict in Europe and the Use of Urban Territories

Brown bears generally avoid urban areas and high-traffic roads, especially in regions with higher population density. This avoidance is stronger in Europe than in North America, likely due to a longer history of human persecution and different species of bears. Most of the problems related to bears in North America are connected with *U. americanus*. While some bears tolerate fragmented landscapes, urbanization remains a major barrier to habitat use and can increase conflict risks. Standardized data on relative and absolute habitat use are needed to better understand and manage bear adaptation to urban environments [8].

Although brown bear conservation is mandated across the EU, its effectiveness varies by country. Romania’s Central Region has the largest bear population, the highest number of damage incidents (8252), and the highest associated costs. In contrast, Croatia reports the fewest damages (113), while Sweden has fewer bears (approximately 2980) and less conflict (762 incidents), despite having a larger habitat. These differences can be attributed to how each country implements management plans and the effectiveness of institutional involvement in compensation systems [23].

Conflicts between brown bears and humans are increasing globally and causing social and economic harm. The ten main mitigation strategies focus on education and physical barriers; however, their success depends on adaptation to local contexts. Sharing effective practices, especially those from North America, is crucial [24]. These conflicts stem from bears’ opportunistic foraging, leading to livestock losses, crop and property damage, and intrusions into settlements. Reactive measures, such as aversive conditioning or translocation, often become ineffective once the bears become accustomed to them. Preventing access to human food sources and fostering community-based, adaptive management are key [25]. Diversionary feeding shows mixed results and is highly context-dependent [26].

Human–brown bear conflicts occur in several countries within the Dinaric-Pindos region. These countries include Slovenia, Croatia, Bosnia and Herzegovina, Montenegro, Serbia, North Macedonia, Albania, and Greece. The conflicts primarily involve livestock depredation, crop damage, and encounters near human settlements [27].

In Bulgaria, human–brown-bear conflicts include livestock loss, crop damage, and an ineffective compensation system [18]. In Greece, brown bear damage near settlements primarily affects small crops, apiaries, and young livestock. Most conflicts occur from May to October. Although overall losses are low, the economic impact can be significant [28].

Human–brown-bear conflicts in Romania have escalated due to ecological and socio-political factors. The country is home to a significant portion of Europe’s brown bear population and has experienced increased bear activity near settlements, resulting in property damage and livestock losses. Key drivers include habitat fragmentation, food scarcity, and unregulated waste, all of which attract bears [19]. However, Romania’s decision to resume brown bear hunting lacks scientific grounding and fails to address key conflict drivers [29].

In Sweden, bear–human conflicts are minimal and well-managed, primarily involving beehives and livestock. Norway reports fewer but more contentious conflicts, primarily due to sheep predation. There is strong regional opposition despite the small bear population. Finland experiences moderate levels of conflict, including crop and beehive damage, but promotes coexistence through preventive measures and broad public support [30].

### 1.3. Brown Bear in Lithuania

According to paleo-zoological data, brown bears were established in the Baltic region at the beginning of the Holocene [31]. In Lithuania, brown bears continued to appear sporadically from the Upper Paleolithic to the Neolithic periods, typically in low proportions. This suggests they were rarely hunted and played a minor role in human subsistence [32]. Bear bones of various ages have been found in the cultural layers of archaeological sites dating from the 5th–1st centuries BCE to the 13th–18th centuries CE [31].

As the human population and number of livestock increased, bears were persecuted for preying on livestock, contributing to their decline. Extensive deforestation during the 17th and 18th centuries further degraded their habitat. By the late 19th century, bears inhabited only the most heavily forested regions. The last bear in Lithuania was reportedly hunted in 1883, though there were occasional sightings throughout the 20th century [31]. In the 21st century, the number of sightings began to increase [3], especially after 2023 [33]. Since 2019, brown bears have been wintering in Lithuania [3], and in 2025, a cub was registered [34]. On 14 June 2025, a young brown bear wandered into the capital of Lithuania, Vilnius [1,2].

The brown bear is included in the Red Data Book of Lithuania under category NA, meaning the taxon is not suitable for evaluation at the regional level because the regional population consists of very few individuals [3]. Consequently, the species is not included in the Red List.

Given the presence of brown bears in the urbanized territory of Lithuania and the increasing number of bear records in the country, we aim to analyze population recovery and the shortcomings of institutional policies and public information in critical situations when the animal poses a real threat. Our main objective is multifaceted, focusing on the intersection of ecological recovery, institutional responses, and public communication. Our core narrative explores how these elements became intertwined when a bear unexpectedly visited Vilnius.

## 2. Materials and Methods

### 2.1. Study Site

The study area covered the territory of Lithuania (Figure 1A), a small country in Northern Europe, characterized by a predominantly flat landscape. The country area is 65,284 km^2^, the human population size was 2,885,891 in 2024, and the population density was 44.2 inhabitants per km2 [35]. Agricultural lands cover 52.6% of the country, forests 33.2%, built-up territories 3.64%, and roads 1.61% [36,37].

With a population of over 573,000, Vilnius is the capital of Lithuania and covers 401 km^2^. It is one of the greenest capitals in Europe, with nearly 40% of its territory covered by forests and parks (Figure 1B). As of 2024, 20% of Vilnius was occupied by buildings, 21% by agricultural land, 2% by water bodies, and 4% by roads [38].

Lithuania is a parliamentary republic, governed by a parliament (further, the Seimas) and headed by the President. The Seimas is chaired and represented by the Speaker of the Seimas. The institutional context of Lithuania for this study is as follows (Table 1).

The legal framework of the social and conservation landscape of Lithuania is based on the EU Birds and Habitats Directives, administered by the Ministry of Environment. The protected areas (strict nature reserves, national parks, reserves, biosphere and Ramsar sites, etc.) cover about 18% of the country’s territory. The legacy non-governmental organizations, e.g., Green Movement, Baltic Environmental Forum, and Ornithological Society, take active part in forest management, sustainable land use, issues related to climate change, habitat restoration, and threatened species conservation.

### 2.2. Information Sources

We collected relevant scientific publications on bears concerning their presence in urban and peri-urban habitats, conflicts with humans, and related topics by searching Google Scholar and Web of Science. We also used Undermind (https://www.undermind.ai/#overview, accessed on 20 June 2025) to search for references.

Information about the brown bear’s visit to Vilnius was only available on news portals: Delfi (https://www.delfi.lt/, accessed on 21 June 2025), LRytas (https://www.lrytas.lt/, accessed on 21 June 2025), TV3 (https://www.tv3.lt/, accessed on 21 June 2025), 15 min (https://www.15min.lt/, accessed on 21 June 2025), and Bernardinai (https://www.bernardinai.lt/, accessed on 21 June 2025). These sources are all included as references.

Data on brown bear records in Lithuania were extracted from the Lithuanian Mammal Atlas [39] until 1998, the Red Data Book of Lithuania [3] until 2020, and the citizen science project database on large carnivores from 2012 to 2018. This database is not available online. Data on brown bear records after 2021 are based on an initiative of the Lithuanian Hunters and Fishers Association [33].

## 3. Results

### 3.1. Brown Bear Population Recovery in Lithuania

From 1975 to 1997, brown bears were recorded ten times in nine Lithuanian districts, primarily in northeastern and southern Lithuania (Figure 2A). The animals usually invaded from neighboring districts in Latvia and Belarus [39]. Between 1990 and 2000, brown bears were detected approximately ten times; however, they were detected more than thirty times between 2015 and 2019, primarily in northeastern, eastern, and southern Lithuania (Figure 2B). There were two known cases of hibernation in 2017–2019 [3].

A brown bear visited Vilnius for the first time in 2003. It was spotted in the northern part of the city, where forests are interspersed with individual gardens. The next year, the same or a different bear was spotted 20 km north of Vilnius [40]. Due to the limited development of mass media at the time, the few newspaper reports that are available did not attract much public attention.

Based on data from Ref. [33], the number of brown bear records increased after 2020: one in 2021, two in 2022, four in 2023 (Figure 2C), nineteen in 2024 (Figure 2D), and thirty-seven as of 19 July 2025 (Figure 3A). Once again, most of the records were in NE, E, and SE Lithuania. All of the individuals, except for one juvenile recorded on 28 March 2025, were adults. If they were registered from identified footprints, their age was considered unknown.

At least three brown bears were present in Lithuania simultaneously in 2025. One individual was recorded several times by wildlife cameras near the city of Kaunas in the central part of the country between 20 May and 17 June [33,41,42,43,44]. The second, an adult, was recorded in the southern part of the country from 4 June to 6 (Figure 3A). The third travelled from the north toward Vilnius, with records on June 6 and 9 [33].

### 3.2. Brown Bear Visit to the Capital City of Lithuania

On June 14, a two-year-old female bear was seen running through the northern outskirts of Vilnius, specifically the areas of Avižieniai and Tarandė (Figure 3B). This sighting was posted in several Facebook groups.

From the night of June 14 to 15, a bear came to the city. It was scared away from the busy street [1] using cars and later visited the central part of the town, close to the two biggest shopping malls (Figure 3B), as well as the Fabijoniškės residential area. News of the bear spread rapidly via internet portals. Since it was Sunday, there was no institutional participation.

The following evening, June 15, the animal moved north and was spotted at around 8 p.m. in the yards of homes in the village of Riešė, located just outside the capital city of Vilnius. Although it was tracked, the animal fled into the nearby forest and escaped capture. The following day, June 16, it was spotted farther north in the village of Purnuškės. On the 17th, the bear was seen near the town of Dubingiai; on the 18th, it was seen approaching the border of Lithuania’s northeastern forests near Pabradė (Figure 3B). The animal’s movements were constantly updated on several internet news portals [45,46]. There were no reports of damage caused by the animal or the animal attempting to feed in residential areas.

We requested hair samples for further analysis in case the bear needed to be sedated and transported. However, the bear left the city on its own. Its feces were not found.

### 3.3. Institutional Response

Some aspects of the institutional response are unclear because the positions presented in the press differ. The responses are presented in a timeline format:June 15 (AM, LMŽD)**.** The AM presented “invasive plans”—first, they would attempt to sedate the animal, tag it with a GPS device, and release it back to the forest; only in a critical case, they would shoot the bear. It was emphasized that lethal measures would only be taken in threatening situations. Hunters may be issued permits, but this would require communication mechanisms at the state level [47].June 16 (AM, VS)**.** Residents were urged to avoid outdoor activities, keep their children indoors, and steer clear of forests. They were also reminded to call 112, the emergency number, if a bear was spotted. Gaps in communication were confirmed and were planned to be addressed [48].June 16 (AAD, LMŽD)**.** It was reported that the bear had left Vilnius. Drones were used to monitor the animal’s movements, which emphasized vigilance and readiness to track its further movements. [49].June 17 (AM, AAD, VS, VrS, LGGC). The deputy minister acknowledged shortcomings in communication and promised to address them by creating action plans for bear-related incidents. They presented a plan that included GPS tracking, sedation, and, in critical cases, shooting. Clear protocols were to be developed. Municipal representatives complained about the lack of information [50].June 17 (LMŽD). The LMŽD and the hunters obtained permission and decided not to shoot the bear, but rather to observe and tag it. They criticized the authorities for suggesting shooting instead of tagging. They emphasized the need for a long-term wildlife management system [51].June 17 (AM, AAD). The deputy minister acknowledged the public’s lack of information regarding the animal’s whereabouts and institutional actions. The challenges of drone surveillance were discussed, as were plans to improve the communication and coordination of operations [52].June 18 (AM, Committee on Environment Protection). The deputy minister explained that shooting was only one of the options considered, and that it would be used only in extreme cases. He emphasized that the primary goal was to allow the bear to return to the wild on its own and that hunters would only intervene as a last resort. Therefore, there were no direct orders to shoot [53].June 18 (AM, Speaker of the Seimas S. Skvernelis). Seimas Speaker S. Skvernelis offered a critical assessment of the situation, calling it “tragicomic” and expressing dissatisfaction with how the institutions handled the situation and communicated [54].June 18 (Seimas members, Committee on Environment Protection). Members of the Seimas were outraged that the authorities intended to shoot the bear too quickly without first taking adequate measures to observe and tag it. There was criticism regarding the inadequate response and lack of transparency [55].June 18 (AM, President of the Republic of Lithuania). The President asked why the authorities did not take clear action, such as tracking, tranquilizing, and tagging the bear, instead of shooting it. His criticism highlights the authorities’ shortcomings in planning and communication [56].

In short, AM and AAD proposed a package of measures ranging from monitoring to shooting (plans A through C). They later acknowledged communication shortcomings and promised to develop clear action algorithms. Hunting organizations sought a more humane approach to marking and monitoring the bear and criticized the threat of shooting without clear criteria. Seimas and the President criticized the authorities’ response, demanding accountability and systemic solutions. Residents and public figures demanded instructions for locals and called for stronger animal protection. They also expressed dissatisfaction with the prospect of hasty shooting. LGGC was recognized as a reliable and responsible partner in wildlife crisis operations and performed central organizational, logistical, and educational functions. LGGC earned praise from journalists for its humane approach and competence.

## 4. Discussion

The discussion expands on the chronological and institutional details presented in the Results section and connects local events to broader European experiences. It emphasizes the symbolic and cultural significance of the bear’s presence.

### 4.1. Urban and Peri-Urban Large Carnivores Pose a Dilemma for Their Management

Brown bears inhabit urban and peri-urban areas in many countries [4,5,6,7,8]. Conflicts with humans are common, particularly in North America and Europe [13,14,15,16,17,18], but not exclusively. In Japan, for example, brown bears are increasingly entering the city of Sapporo due to population recovery, expanded green spaces, and increased food sources. Though encounters are rare, media-driven fear fuels support for lethal control. Human–bear conflicts reflect the challenges of conserving wildlife in biodiverse cities where public attitudes vary and experience with wildlife is limited [57]. Studies have shown that urban planning and waste management can reduce conflicts and support coexistence between humans and black bears in Alaska, which face similar problems [58].

Though not fully urban-dwelling, these apex predators benefit from their proximity to cities. Understanding how large carnivores adapt to urbanization, including their ecology and behavior, offers crucial insight into managing and conserving these species amid widespread habitat loss [4].

In Greece, effective strategies such as electric fencing, guard dogs, improved herd management, and linking compensation payments to preventive actions have mitigated damage caused by brown bears [28].

A study in Bulgaria emphasizes the necessity of improved data collection and coordinated management, as well as proactive measures such as electric fencing and public awareness campaigns, to mitigate human–bear conflicts and achieve long-term conservation objectives [18].

Inconsistent implementation of compensation schemes and controlled bear removal in Romania has led to a lack of public trust. To effectively mitigate conflicts, a more balanced strategy is advised, including preventive measures such as electric fencing, improved waste management, and better institutional coordination [19]. Romania’s bear hunting policy risks exacerbating human–bear conflicts and encouraging trophy hunting. A science-based management approach, public education, and improved urban coexistence strategies are urgently needed instead [29].

Recent reports have highlighted the politicization of bear encounters in Romania and Slovakia. There have been debates over population estimates, EU protections, and proposals for large-scale culls to address safety concerns [59]. Similarly, in southeastern Poland, a mayor is pushing for urgent changes to hunting laws to allow for the selective culling of brown bears amid a surge in human–bear encounters. This underscores the tension between public safety and conservation priorities [60].

In Sweden, brown bears avoid areas near towns and recreational resorts. Over 74% of female bears prefer rugged terrain more than 10 km from human settlements. Bears found closer to urban areas tend to be significantly younger, mainly sub-adults, suggesting these zones are used by dispersing individuals rather than established adults. As resorts expand near protected areas, they risk fragmenting bear habitat and hindering recolonization. This highlights the need to preserve undeveloped, rugged corridors for successful urban-edge bear conservation [61].

In summary, the space use of brown bears varies widely and is strongly influenced by the human footprint, vegetation productivity, and forest disturbances. Bears used smaller ranges and moved less in human-dominated or resource-rich areas. Conversely, forest cover encouraged movement. Protected and roadless areas had little effect. Human-altered landscapes may limit the expansion and connectivity of bear populations, underscoring the importance of maintaining forest integrity to support viable populations [62]. Brown bears in Fennoscandia exhibit strong avoidance behavior regardless of proximity to roads or settlements [63]. Roads have a minimal impact on overall habitat suitability with no widespread road avoidance, except among females with cubs during denning [64]. In 2024, a brown bear was hit by a train in northern Lithuania [65], and a young female bear that arrived in Vilnius had to cross busy roads several times (see Figure 3).

### 4.2. Lithuanian Brown Bears: Visitors Stay?

The distribution of brown bear records in the country (see Figure 2 and Figure 3) indicates that the animals are coming from two neighboring countries, Latvia [66,67] and Belarus. However, bear movement to and from Belarus is currently complicated by the cutting of wire fence and wall barriers on the border, which were finished in August 2022. The total length of the Lithuanian–Belarusian border is approximately 679 km, of which about 502 km is covered by a physical barrier, while the remaining sections (about 100 km) are difficult to access due to rivers and lakes [68]. The entire border is covered with video surveillance and IT systems.

Therefore, bears can only enter Lithuania from the south via swamps [69] or bodies of water that are not covered by the barrier [70]. However, the barrier also limits their ability to leave the country. In 2023, bears were observed walking several kilometers alongside the border fence [71] or even attempting to climb it [72].

A more accurate assessment of the origin of Lithuanian bears requires genetic testing and analysis of animal movement via transmitters. Without this information, it can only be concluded that the bear population in Lithuania is recovering.

### 4.3. Lithuanian Experiences: The Moose and the Brown Bear Visit to the Capital City

Three weeks before the brown bear’s visit, on 21 May 2025, a moose (*Alces alces*) visited the capital city of Lithuania. It was spotted wandering around the city center and several other streets from morning until evening. Police and rescue services followed the animal using flashlights, sirens, and firecrackers [73]. Finally, the moose was driven into the river [74]. After crossing the river, the moose escaped and left the city. It was extremely stressed and frightened by the crowds of people photographing it and trying to scare it away. The incident revealed a lack of clear public communication on the part of the authorities—residents did not know how to behave, and some got too close. Naturalists argued that a plan should be prepared for such cases, involving professionals who would monitor the animal and avoid causing additional stress to either the animal or people [75].

Do residents of Vilnius feel safe in a city where wild animals roam the streets? A one-question survey of 1200 residents revealed that nearly 26% do not feel safe and believe special services should do more to ensure their safety. Another 13% said they started to feel uncomfortable. However, 57% said they feel safe and believe the bear should be left alone. The remaining 4% said this issue did not concern them [76]. Therefore, the discourse journalists used in the bear case was possibly different. The tone of communication from institutions was predominantly critical, noting their failure to provide timely and clear information. Particular emphasis was placed on the decisions made by hunters, who were praised for their humanity, as well as for their marking and monitoring efforts. Political reactions (Skvernelis, Nausėda, and the Seimas) are presented as harsh and demanding change. While most of the texts are informative and analytical, they also offer sharp political and social criticism of bureaucratic shortcomings, hasty decisions, and unclear plans.

The bear’s appearance in Vilnius aroused curiosity and revealed serious systemic problems and was, therefore, exploited by various groups. Experts warned that the bear could cause property damage, such as damaging cars, beehives, or crops. Residents were advised to check if their insurance covered damage caused by wild animals [77]. While the bear wandered around the city, the Deputy Minister of the Environment admitted that he was enjoying his personal time during the event. He stated, “We were barbecuing, like normal people.”, thus drawing criticism from the public and politicians for his inaction and lack of institutional coordination [78,79]. The bear was later tracked by drones, but its movement toward the city indicates that monitoring measures are ineffective. Despite the seriousness of the situation, some companies used the bear for advertising and social media content, creating witty campaigns that attracted public attention [80]. Three days later, when the bear was spotted 50 km away from Vilnius [81], the discussions and comments ended.

Lithuanian readers expressed disbelief in the authorities’ ability to respond quickly and effectively. At the same time, they supported the hunters’ initiative to resolve the situation humanely. Sarcastic remarks reflect public dissatisfaction with how such situations usually devolve into a “circus” rather than an orderly nature conservation operation. Readers criticized delays, unclear communication, and “forgotten” actions. The hunters’ refusal to shoot was met with approval and respect for their humanity. These comments encourage rethinking the competence of institutions and their preparedness for similar crises in the future.

### 4.4. Foreign Media Framing of the Vilnius Bear Incident

Social media has the potential to be a powerful tool for biodiversity conservation. It can spread awareness, encourage pro-conservation behavior, boost funding, and influence policy. However, it also poses risks, including promoting species exploitation, illegal wildlife trade, unsustainable tourism, and misinformation [82].

In Manitoba, Canada, most residents adjusted their behavior to avoid large carnivores based on social media posts. Despite limited formal training, they were hesitant to contact authorities due to mistrust and fear of lethal action [83]. Similarly, social media videos in Tibet revealed that most encounters with carnivores, such as snow leopards and bears, were neutral. Negative interactions were linked to proximity, species type, and the presence of other animals. These studies underscore the value of social media in documenting local human–wildlife dynamics and supporting participatory conservation [84].

During the first four days, the brown bear issue was covered by The Guardian (UK), the Associated Press (AP News, United States), TVP World (Poland), and UNN.ua (Ukraine). Their coverage generally took a positive tone, ignoring the more critical texts in the local mass media.

Readers were informed that, after a brown bear appeared in Vilnius, the Lithuanian authorities issued a permit allowing the animal to be lethally removed if it posed a threat. However, the permit was precautionary, not an explicit order to kill. The public widely misunderstood this nuance, leading to concern and criticism of the government’s approach [85,86].

The Lithuanian Association of Hunters and Fishermen publicly refused to act on the permit, asserting that the bear was not aggressive and did not warrant lethal control. This ethical stance received positive media coverage, especially considering the species’ rarity in Lithuania, where only five to ten individuals are estimated to remain [86,87]. The media framed this outcome as a successful example of nonlethal human–wildlife conflict mitigation, reinforcing the potential of coexistence-oriented strategies [85,86].

In response to the incident, Vilnius city officials launched a drone-based wildlife monitoring program. The drones, which are equipped with thermal cameras, are intended to help detect and manage future incursions of wildlife into urban areas without resorting to lethal measures immediately [88].

### 4.5. Symbolic Wildlife and the Media: The Bear as Spectacle, Distraction, and Narrative Device

A bear in the streets of Vilnius is a rare and extraordinary event that sparks the collective imagination. The bear symbolizes wild nature, danger, and naturalness. Thus, its appearance in the city is emotionally charged news. Media outlets exploit such symbols because they resonate with the public’s feelings, fears, and nostalgia [89,90].

According to J.G. Webster and T.B. Ksiazek, the economics of media attention [91] explains increased local media interest in the bear visit. Mass media operate according to the logic of attracting readers. Not everything important receives attention. Rather, attention is given to what is distinctive, unexpected, or visually striking. Unlike complex issues such as tax reform or geopolitical threats, which require more explanation and do not elicit an immediate emotional response, the bear’s visit is easily broadcast as “front-page news.” Large carnivores are generally “attention-drawing” species [92].

Such publications can act as a psychological buffer during periods of geopolitical tension or financial strain (e.g., tax increases in Lithuania), when society experiences collective stress. Consciously or unconsciously, the media offers the theme of “distraction” or diversion [93], and the bear becomes a symbolic escape from tension, as a “lighter” current event.

The bear also reflects the narrative of the relationship between humans and nature [94]. The presence of a bear in the city raises questions about urbanization, climate change, and the return of wildlife to urban areas. These topics enable the media to discuss environmental protection and urban development in a non-politicized manner.

In summary, the profuse coverage of bears in the local media can be understood through a sociological lens as an agenda-setting strategy, a symbolic expression of collective anxiety, and a subconscious search for emotional balance.

## 5. Conclusions

The Vilnius bear incident exemplifies the intersection of urban wildlife encounters, media narratives, public emotions, and ethical debates. The bear became a symbol of the wilderness, evoking fear and fascination, as it was framed by institutional responses and societal tensions. The refusal to implement lethal control and the use of drone surveillance reflect a shift in norms toward coexistence and technological monitoring. This case highlights the importance of considering the social and symbolic aspects of managing large carnivores at the urban–wild interface.

Alongside the shift toward nonlethal coexistence, urban wildlife management must ensure that all stakeholders can shape the decision-making [95]. Proactive, nonlethal management that combines drone surveillance, GPS tagging, clear communication protocols, and cross-institutional coordination can safely guide recovering brown bears out of urban areas and foster lasting human–wildlife coexistence.

## Figures and Tables

**Figure 1 animals-15-02151-f001:**
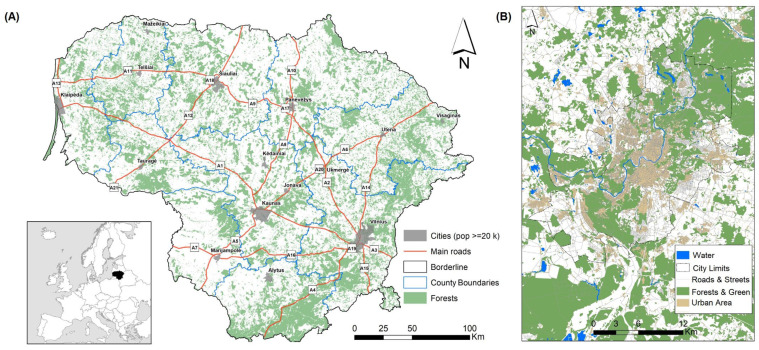
Map of Lithuania with indicated cities, main roads, and forests (**A**), and map of Vilnius city and suburbs (**B**). Source: Public Institution Construction Sector Development Agency. NZT: Geographic data services. Lithuanian Spatial Information Portal (Geoportal). Available online: https://www.geoportal.lt/arcgis/rest/services/NZT (accessed on 18 June 2025).

**Figure 2 animals-15-02151-f002:**
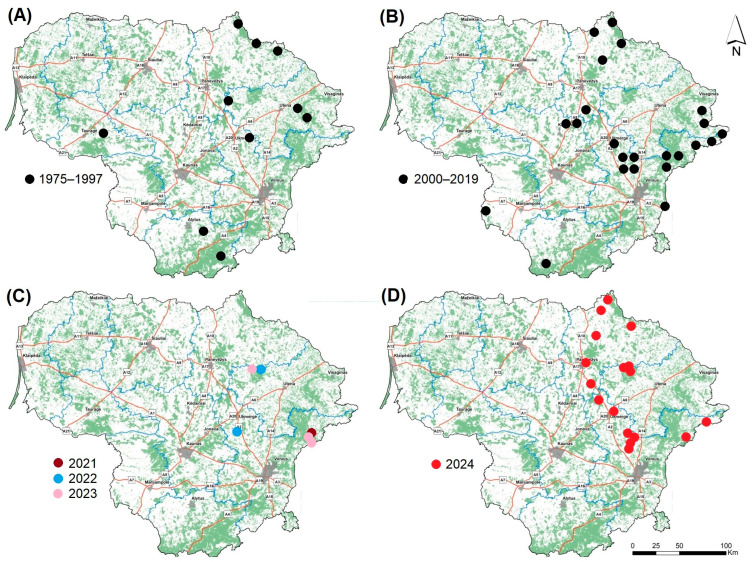
Brown bear records in Lithuania (**A**) 1975–1997, (**B**) 2000–2019, (**C**) 2021–2023, and (**D**) 2024.

**Figure 3 animals-15-02151-f003:**
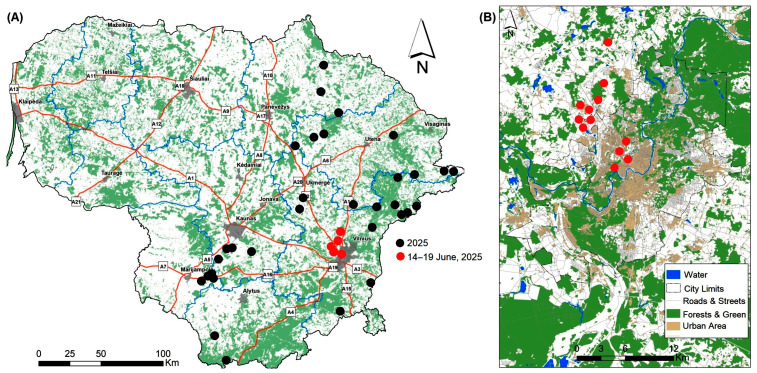
Brown bear records in Lithuania in 2025 (**A**), outskirts of Vilnius, 14–19 June 2025, and Vilnius city, 15 June (**B**).

**Table 1 animals-15-02151-t001:** Institutions in Lithuania’s environmental governance, as related to this study, and their core functions. Abbreviations are based on Lithuanian language.

Name (Abbreviation)	Functions
Ministry of Environment (AM)	Develops national conservation policy and international agreements.
Environmental Protection Department (AAD)	Subordinate to AM; enforces environmental laws, issues permits, and conducts inspections.
Vilnius municipality (VS)	Implements environmental issues, including the maintenance of biodiversity and green spaces, as well as community outreach at the city level
Vilnius district municipality (VrS)	Implements environmental issues and community outreach at district level.
Lithuanian Hunters and Fishers Association (LMŽD)	Represents hunting and fishing interests; conducts field operations such as roadkill cleanup.
Wildlife Rescue Center (LGGC)	A subsidiary of the Lithuanian University of Health Sciences; cares for, rehabilitates, and transports injured or distressed wild animals. Located about 115 km from the center of Vilnius.

## Data Availability

The original contributions presented in this study are included in the article and links to webpages. Further inquiries can be directed to the corresponding authors.

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
