# Peer review of "A Brown Bear’s Days in Vilnius, the Capital of Lithuania"

_animals, 2025, doi:10.3390/ani15142151_

Round 1

Reviewer 1 Report

Comments and Suggestions for Authors

I think the topic is important and the topic of interest to those interested in urban wildlife concerns. However, I think the author/s need to rethink the manuscript and revise due to a lack of clarity.

  1. There is proofreading and editing to be done; most of the issues are related to English as a second language, so getting an English speaker to do a copy edit will fix problems quickly.
  2. A lot of the references are not in English so I cannot assess their appropriateness; it might be a lot to ask, if so ignore this, but a rough English translation of titles would be helpful. Again, a lot of work so I leave it to the editors to make a call on this.
  3. The biggest problem I have with the manuscript is that, in the end, I am not sure what the author/s are trying to demonstrate. Is the main goal to assess bear conservation in Lithuania? Is it to understand public perceptions of potentially dangerous wildlife? Is it to assess a government's response (and failures) around addressing the bear? Or is the goal to discuss public responses to the bear and the uses of social media to influence perceptions? All of these topics are here but they do not hang together coherently. I don't even know if the bear in question lived. I think the main argument you want to make needs to be clearly identified and then results and discussion fitted to that. Similarly, there is a very good literature review, but it is a little thin on the public perception side and that seems to be the direction the manuscript is trying to go in. Any of the topics I've listed would be fine, they are all there in the manuscript, but I personally, the author can decide, would suggest continuing the story of the bear in the capital and using that to more clearly focus on perceptions of the officials and the public. 
  4. While the above is the biggest issue, if addressed should allow the following to be resolved: the results section does not link to the discussion. The results are mostly a blow by blow account of how the government communicated the bear's presence and their actions. The discussion is much broader, does not draw upon the results particularly and brings in new data (which a lot of academics frown upon). But again, this goes back to what your main argument is. If you wish to dissect the government's actions, with rewriting (not, I suggest, direct quotes) the description of government's actions are fine, but then your discussion needs to focus on that topic. If you want to discuss public perceptions and actions (which is where the discussion seems to want to go) then the results need to focus on the data you are analyzing, a great deal of which gets brought in to the discussion.
  5. In short, figure out what you think is really interesting and then present that finding. It is there, you are just trying to cover too much in an unorganized way. I think you can keep much of what you want to share, but fit it to the narrative you clearly describe.
  6. Let us know if the bear made it.
  7. I really hope you will revise and resubmit, there is a really interesting piece of research here that I think is important to share. 
Comments on the Quality of English Language

Needs proofreading to correct English grammar issues.

Author Response

Rev #1 comments and answers

I think the topic is important and the topic of interest to those interested in urban wildlife concerns. However, I think the author/s need to rethink the manuscript and revise due to a lack of clarity.

Comment 1: There is proofreading and editing to be done; most of the issues are related to English as a second language, so getting an English speaker to do a copy edit will fix problems quickly.

Answer: thank you for leaving final copyediting and proofreading to Editors, on our side we re-read this text, with consulting to native speaker. We present also a “clean” version.

Comment 2: A lot of the references are not in English so I cannot assess their appropriateness; it might be a lot to ask, if so ignore this, but a rough English translation of titles would be helpful. Again, a lot of work so I leave it to the editors to make a call on this.

Answer: Sure, we add translated titles in square parentheses for all non-English references. Please understand that these references are not available in English.

Comment 3: The biggest problem I have with the manuscript is that, in the end, I am not sure what the author/s are trying to demonstrate. Is the main goal to assess bear conservation in Lithuania? Is it to understand public perceptions of potentially dangerous wildlife? Is it to assess a government's response (and failures) around addressing the bear? Or is the goal to discuss public responses to the bear and the uses of social media to influence perceptions? All of these topics are here but they do not hang together coherently. I don't even know if the bear in question lived. I think the main argument you want to make needs to be clearly identified and then results and discussion fitted to that. Similarly, there is a very good literature review, but it is a little thin on the public perception side and that seems to be the direction the manuscript is trying to go in. Any of the topics I've listed would be fine, they are all there in the manuscript, but I personally, the author can decide, would suggest continuing the story of the bear in the capital and using that to more clearly focus on perceptions of the officials and the public. 

Answer: We formulated our aim in this form (Lines 130–133) – “Based on the fact of brown bear presence in urbanized territory of Lithuania and the increase of number of bear records in the country, we aim to analyze the recovery of the population and the shortcomings of institutional policy and public information in an unexpected (critical) situation when the animal could pose a real threat.”

Our main objective was indeed multifaceted. We sought to illustrate institutional perceptions and responses regarding the unexpected presence of a brown bear in Vilnius, Lithuania, exploring how this event could influence conservation discourse and urban wildlife management practices. Specifically, our intention was to analyze how social media, public opinion, and institutional responses intersected during this high-profile wildlife encounter.

However, we cannot sharpen narrative around public perceptions, delineating the interplay between media representation, public reaction, and institutional responses, as public perceptions were not investigated, except of a one-question survey. To address your comment on the public perceptions, we accessed survey results again, and expanded text at Line 336. “Do Vilnius residents feel safe in a city where wild animals roam the streets? A one-question survey of 1,200 residents showed that nearly 26% do not feel safe and believe that special services should work better to ensure their safety. Another 13% started to feel uncomfortable. However, 57% of respondents said they feel safe and that the bear should be left alone. The remaining 4% replied that this issue did not concern them.”

Comment 4: While the above is the biggest issue, if addressed should allow the following to be resolved: the results section does not link to the discussion. The results are mostly a blow by blow account of how the government communicated the bear's presence and their actions. The discussion is much broader, does not draw upon the results particularly and brings in new data (which a lot of academics frown upon). But again, this goes back to what your main argument is. If you wish to dissect the government's actions, with rewriting (not, I suggest, direct quotes) the description of government's actions are fine, but then your discussion needs to focus on that topic. If you want to discuss public perceptions and actions (which is where the discussion seems to want to go) then the results need to focus on the data you are analyzing, a great deal of which gets brought in to the discussion.

Answer: As said in the previous answer, our main objective was indeed multifaceted, therefore, it also defined number of issues to be covered in Discussion.

MDPI says: “The findings and their implications should be discussed in the broadest context possible”. Therefore, we organize Discussion in the way (i) to understand global and European context, comparing Lithuania’s situation with international experiences; (ii) to understand national context by analyzing Lithuanian historical and recent trends in bear appearances; (iii) to present institutional and public reactions: in the frame of media narratives, analyzing both local and international media coverage and symbolic implications.

Comment 5: In short, figure out what you think is really interesting and then present that finding. It is there, you are just trying to cover too much in an unorganized way. I think you can keep much of what you want to share, but fit it to the narrative you clearly describe.

Answer: We recognize that our approach covers multiple topics simultaneously without highlighting a central, most interesting finding: the bear’s presence in Vilnius as a lens through which population (re)establishment in the country, institutional reactions, and media narratives intersected. Each section aligns around this central narrative thread.

We present Results organized into three sections: Population recovery (historical overview and recent sightings, using maps for illustration), then Bear visit to Vilnius (chronological narrative of events related to the bear sighting and subsequent movements), and finally, Institutional response (detailed, clearly structured, and date-stamped timeline of institutional actions, reactions, and public criticisms).

Therefore, we would prefer to maintain our current format and narrative. We believe this approach effectively captures the complexity and interconnectedness of ecological, institutional, and societal responses associated with the bear’s visit to Vilnius.

Comment 6: Let us know if the bear made it.

Answer: O yes, she did. This was said in Line 219 and referred in [49]. We add additional reference to prove, that bear left Vilnius and now is safe at Line 357

Comment 7: I really hope you will revise and resubmit, there is a really interesting piece of research here that I think is important to share. 

Answer: Please find revised manuscript. We did some changes in the text according your comments 3–5, but hope you will agree with our framing of the situation, even if it is a bit different from your point of view. And, thank you for your comments, they were really helpful.

Reviewer 2 Report

Comments and Suggestions for Authors

We only know that it was a subadult bear. It would be interesting to know whether it was just a lost yearling or 2-3 year old able to survive on its own. Also it would be good to know whether it was a female or a male? Additional information which would be of interest to readers would be some details on bear' behaviour - whether it was trying to feed on garbage, break into some properties, visited peoples' gardens etc. Finally - were there any attempts to collect a fecal sample to have a DNA for future comparisons?

In conclusions it would be good to include a short paragraph on general problems with wildlife (especially large or invasive species) encroaching cities, discussing possible solutions for estimating and controlling their population numbers.

Author Response

Rev #2 comments and answers

Comment: We only know that it was a subadult bear. It would be interesting to know whether it was just a lost yearling or 2-3 year old able to survive on its own. Also it would be good to know whether it was a female or a male? Additional information which would be of interest to readers would be some details on bear' behaviour - whether it was trying to feed on garbage, break into some properties, visited peoples' gardens etc. Finally - were there any attempts to collect a fecal sample to have a DNA for future comparisons?

Answer: it was a two-year-old female bear, information added to beginning of 3.2. chapter, to Simple summary and to Abstract. We apologize this fact was missing in the time of manuscript preparation.

Lines 201–203 indicates bear was running through the residential yards. However, there were no reports of damage caused by the animal or it’s attempting to feed in residential areas. We added this information to the end of 3.2 chapter.

We asked to collect hair for further analysis in the event that the bear was sedated and transported. However, the bear left the city on its own. Its feces were not found.

Comment: In conclusions it would be good to include a short paragraph on general problems with wildlife (especially large or invasive species) encroaching cities, discussing possible solutions for estimating and controlling their population numbers.

Answer: we address your comment adding the last paragraph to discussion, based on paper on “Pathways between people, wildlife and environmental justice in cities”: Although urban wildlife is vital, the impact of wildlife on city residents remains understudied through an environmental justice lens. There are different ways in which the presence of wildlife and its management can exacerbate social inequities, often disadvantaging marginalized communities. Rather than labeling wildlife as "good" or "bad," cities need equitable and inclusive decision-making processes that support healthy wildlife populations and advance social justice [McInturff et al., 2025].

Comment Line 48: there is a number of Romanian reports on such cases (they are cited later) and Comment Line 65: recently several such cases in Slovakia and some in Poland

Answer: we address your comment including two new references in Discussion, at Line 292, with the text “Some recent reports have highlighted the politicization of bear encounters in Romania and Slovakia. Debates have emerged over population estimates, EU protections, and proposals for large-scale culls to address safety concerns [ref]. Similarly, in southeastern Poland, a mayor is pushing for urgent changes to hunting laws to allow for the selective culling of brown bears amid a surge in human–bear encounters, underscoring the tension between public safety and conservation priorities [ref].”

Comment Line 75: and different bear species, most of problems related to bears in North America is connected with black bears

Answer: inserted as recommended.

Comment: record should be used instead of registrations

Answer: requested changes done.

Round 2

Reviewer 1 Report

Comments and Suggestions for Authors

I appreciate the changes you've made. I think it would be improved though better focus, but if the editors feel a broad approach is accrptable, that would be there call.

Author Response

Rev#2 round 2 comments and answers

Comment: I appreciate the changes you've made. I think it would be improved though better focus, but if the editors feel a broad approach is accrptable, that would be there call.

Answer: thank you. We agree that a narrower focus is needed for a deeper analysis. Still, our intention was to provide a broader view. The paper provides a broad discussion for those wanting an overview of the issue. We would be happy to stay with the current manuscript structure.